# A Clinical Prediction Rule for Thrombosis in Critically Ill COVID-19 Patients: Step 1 Results of the Thromcco Study

**DOI:** 10.3390/jcm12041253

**Published:** 2023-02-04

**Authors:** Karen L. Ramírez Cervantes, Elianne Mora, Salvador Campillo Morales, Consuelo Huerta Álvarez, Pilar Marcos Neira, Kapil Laxman Nanwani Nanwani, Ainhoa Serrano Lázaro, J. Alberto Silva Obregón, Manuel Quintana Díaz

**Affiliations:** 1Patient Blood Management Research Group, Hospital La Paz Institute for Health Research, 28040 Madrid, Spain; 2Department of Statistics, Charles III University of Madrid, 28903 Getafe, Spain; 3Department of Public Health & Maternal and Child Health, Faculty of Medicine, Complutense University of Madrid, 28040 Madrid, Spain; 4Intensive Care Unit, Hospital Germans Trias i Pujol, 08916 Badalona, Spain; 5Intensive Care Unit, La Paz University Hospital, 28040 Madrid, Spain; 6Intensive Care Unit, Clinic University Hospital of Valencia, 46010 Valencia, Spain; 7Intensive Care Unit, University Hospital of Guadalajara, 19002 Guadalajara, Spain

**Keywords:** thrombosis, COVID-19, risk prediction model, clinical prediction rule

## Abstract

The incidence of thrombosis in COVID-19 patients is exceptionally high among intensive care unit (ICU)-admitted individuals. We aimed to develop a clinical prediction rule for thrombosis in hospitalized COVID-19 patients. Data were taken from the Thromcco study (TS) database, which contains information on consecutive adults (aged ≥ 18) admitted to eight Spanish ICUs between March 2020 and October 2021. Diverse logistic regression model analysis, including demographic data, pre-existing conditions, and blood tests collected during the first 24 h of hospitalization, was performed to build a model that predicted thrombosis. Once obtained, the numeric and categorical variables considered were converted to factor variables giving them a score. Out of 2055 patients included in the TS database, 299 subjects with a median age of 62.4 years (IQR 51.5–70) (79% men) were considered in the final model (SE = 83%, SP = 62%, accuracy = 77%). Seven variables with assigned scores were delineated as age 25–40 and ≥70 = 12, age 41–70 = 13, male = 1, D-dimer ≥ 500 ng/mL = 13, leukocytes ≥ 10 × 10^3^/µL = 1, interleukin-6 ≥ 10 pg/mL = 1, and C-reactive protein (CRP) ≥ 50 mg/L = 1. Score values ≥28 had a sensitivity of 88% and specificity of 29% for thrombosis. This score could be helpful in recognizing patients at higher risk for thrombosis, but further research is needed.

## 1. Introduction

There is sufficient clinical evidence indicating that coronavirus disease 2019 (COVID-19) is associated with thrombotic complications, increasing disease severity [1,2]. The incidence is exceptionally high in critically ill individuals admitted to intensive care units (ICUs), in whom both venous thromboembolism (VTE) and pulmonary embolism (PE) have been observed in more than 20% of patients, especially during ancestral Delta and Omicron variants, a trend that seemed to decrease with the new variants [1,3,4,5]. In hospitalized individuals, the incidence is greater when assessed according to screening than by clinical diagnosis [6,7]. For instance, when systematic computer tomography pulmonary angiogram is performed in all hospital-admitted patients, higher rates of thromboembolism are observed [8]. However, systematic thrombosis screening is not currently indicated in COVID-19 individuals, and other predictive tools must be developed.

Before the ongoing pandemic, the Geneva and Wells scores were the most used to predict PE and deep vein thrombosis (DVT) in the general population, respectively [9,10]. Still, in COVID-19 individuals, their efficacy has not been proven [11]. Therefore, other predictive scores have been adapted to respond to the need for early thrombosis identification [12,13]. However, their application has been hampered by their low sensitivity and specificity, the use of variables hardly used outside of a few limited settings, and a lack of validation in clinical settings [13].

Early identification of predictive factors for thrombosis could improve clinical decision making to treat and reduce the morbidity and mortality in COVID-19 subjects. Hence, there is a need to systematically assess the risk of thrombosis in hospitalized COVID-19 patients and develop methodical diagnostic protocols. Therefore, the present study aimed to develop a clinical prediction rule for thrombosis in hospitalized COVID-19 population.

## 2. Materials and Methods

### 2.1. Study Design and Ethics

We conducted a cross-sectional retrospective observational study with a clinical prediction rule for thrombosis in hospitalized COVID-19 patients that required ICU admission. To do so, we developed a scoring system based on the recommendations of Zhang et al. [14]. We also considered the Transparent Reporting of a Multivariable Prediction Model for Individual Prognosis or Diagnosis (TRIPOD) guidelines [15].

Our study was approved by the Ethics Committee of La Paz University Hospital.

### 2.2. Source of Data

The present investigation is part of the Thromcco Study Project (TSP), a multicenter retrospective database that contains the de-identified data of hospitalized patients admitted to the ICUs of the following Spanish hospitals: La Paz University Hospital in Madrid, Germans Trias I Pujol Hospital in Barcelona, University Hospital in Guadalajara, University Hospital in Burgos, Parc Taulí University Hospital in Sabadell, Clinical University Hospital in Valencia, Clinical University Hospital in Valladolid, and Son Espases University Hospital in Palma de Mallorca. We managed the data-collecting process by creating the study database in the REDcap clinical data repository, a secure web application for managing hospital databases that provide a standard for data collection among all involved medical institutions. Access to this repository was authorized for the professionals in charge of the data management of every participating hospital, who had at their disposal a database replication-blinded to other hospitals’ information. Only authorized data analysts (KLRC, SCM, and EM) could access all database instances. 

### 2.3. Participants

Consecutive hospitalized COVID-19 patients aged ≥18 years who were admitted to the ICUs of the participating hospitals between March 2020 and October 2021 were studied. All ICU-admitted subjects had a confirmed reverse-transcription polymerase chain reaction (RT-PCR) test positive for severe acute respiratory syndrome coronavirus 2 (SARS-CoV2). Patients were followed from hospital admission (index date) to hospital discharge or death.

### 2.4. Variables

The Thromcco database comprises 478 variables composed of hospital and ICU records collected retrospectively. To perform this study, we selected the following variables: sociodemographic data (age, sex, race, and smoking habit), body mass index (BMI), blood type, previous comorbidities (hypertension, diabetes, obesity (divided into categories as class 1: BMI 30–34.9, class 2: BMI 35–39.9, and class 3: BMI >40), asthma, chronic obstructive pulmonary disease (COPD), and ischemic and valvular heart disease), length of hospital and ICU stays, number of venous doppler ultrasounds of the lower limbs performed, anticoagulant regimen received (prophylactic, intermediate, and therapeutic), blood components transfused (red cells, fresh-frozen plasma, and platelets), and requirements of invasive and noninvasive mechanical ventilation, tracheotomy, or prone positions. We also included the blood test results (D-dimer, fibrinogen, leucocytes, lymphocytes, platelets, ferritin, C-reactive protein (CRP), and interleukin 6 (IL6)), prothrombin time (PT), procalcitonin, creatinine, lactate dehydrogenase (LDH), aspartate dehydrogenase (AST), and alanine transaminase*(ALT)) that were collected at admission and on days 1, 2, 5, and 10 of hospitalization. Adverse outcomes such as sepsis and death were also gathered.

### 2.5. Statistical Analysis and Predictors

The primary outcomes of our study were venous thrombosis, DVT, PE, and catheter-related thrombosis. The secondary outcomes was arterial thrombosis, considered when a stroke or myocardial infarction occurred. Only thrombotic events registered during hospitalization were studied. If patients had more than one admission to the ICU, only the first one was considered.

Patients from the TS database with large proportions of missing data (>30% of selected variables) were excluded. To determine the factors predictive for thromboembolism, samples were randomly split into a training set, including 70% of patients, and a test set, considering the remaining 30%. Diverse logistic regression model configurations were performed, including demographic data, pre-existing conditions, and blood tests collected during the first 24 h of hospitalization. Once we obtained a model with statistically significant predictors (*p*-value < 0.05) and overall accuracy above 70% (training set), this model was validated through computations of accuracy and performance using the remaining 30% of patients (test set). A receiver operating characteristic (ROC) curve was generated in conjunction with the area under the curve (AUC) to assess the discriminative ability of the final model. 

Once the model that better predicted thrombosis was obtained, we developed a scoring system for thrombosis risk stratification following the recommendations of Zhang and colleagues. The numeric and categorical variables included in the model were converted to factor variables, giving a score to the values obtained. This score is named the Thromcco Study (TS) score.

To establish predictive cut-off values, subjects that did not present a thrombotic event during hospitalization were considered the control group (*n* = 60). Thus, considering the prevalence of thrombosis in our sample (20.1%), we calculated the sensitivity (SE) and specificity (SP) of the score and its positive and negative predictive values (PPV and NPV, respectively). In addition, we evaluated the capacity of the TS score as a tool to indicate an imaging test to detect DVT by determining the doppler ultrasounds of the lower limb veins that would be needed to diagnose one case of thrombosis. Only the subset of subjects with a doppler ultrasound was considered for this last analysis.

Categorical variables are reported as count data by frequency, while continuous variables are reported as mean ± standard deviation or median and interquartile range (IQR). Patients’ characteristics were compared between subjects with and without thrombosis using chi-square or Fisher’s exact test (categorical variables) and Mann–Whitney or Kruskal–Wallis tests (continuous variables), setting the significance level to 0.05.

The statistical analysis of this study was performed in R version 4.1.3 (17 March 2022).

## 3. Results

Out of 2055 subjects with COVID-19 registered in the Thromcco database, only 299 patients were considered for the final TS score development and analysis. This final data subset resulted from diverse model configurations that only included patients with complete medical records.

### 3.1. Patient Characteristics

The median age of participants was 62.4 years (interquartile range (IQR), 51.5–70), and most of them were men (79.9%). Hypertension (40.8%), obesity (35.8%), and diabetes (22.7%) were the most common chronic comorbidities at baseline (at COVID-19 diagnosis). Subjects were hospitalized for a median of 28 days (IQR 18–42 days), of which 14 (IQR 7–28 days) stayed in the ICU. The median time from hospital admission to ICU admission was two days (IQR 0–5 days). Blood test results during the first 24 h of hospital admission showed elevated median levels of D-dimer (1676, IQR 779–4084), fibrinogen (719, IQR 608–861), ferritin (974, IQR 482.1–1634), CRP (126, IQR 69.6–207.6), PT (12.9, IQR 11.9–15.6), and IL6 (71.3, IQR 36.5–167.8). During ICU admission, 70.9% of subjects required invasive mechanical ventilation (IMV) and 47.2% required a tracheotomy; during the whole hospitalization, 15.4% developed sepsis, and 29% died due to COVID-19 (Table 1).

The incidence of thrombosis was 20.06% (*n* = 60). DVT accounted for 78.3% of cases (*n* = 47), of which 32% (*n* = 15) also presented a PE. Compared with the control group (*n* = 239), subjects with thrombosis were older (63.2 years (IQR 53–72) vs. 60 years (IQR 51–65.9), *p* = 0.043), had more extended hospital and ICU stays (35.5 days (IQR 25–53) vs. 27 days (IQR 17–37) in hospital, *p* = 0.013; and 27.5 days (IQR 15–40) vs. 12 days (IQR 7–24) in the ICU, *p* = 0.001), needed more blood and platelets transfusions (50% vs. 25.5% *p* = 0.000; and 13.3% vs. 5.4%, *p* = 0.013, respectively), and more commonly developed sepsis (33.3% vs. 17.9%, *p* = 0.002). Moreover, in the ICU, they required more IMV (88.3% vs. 66.5% *p* = 0.001), tracheotomy (60% vs. 43.9%, *p* = 0.028), and prone positions (81.6% vs. 53.5%, *p* = 0.000). Furthermore, without statistical significance, the mortality rate was higher in patients with thrombosis than in those without it (38.3% vs. 26.7%, *p* = 0.078) (Table 2).

### 3.2. Risk Prediction Model

The model showed, with an SE of 83% and an SP of 62%, that age; sex; levels of D-dimer, leucocytes, and IL6 collected at admission; and levels of CRP collected during the first 24 h of hospitalization could predict thrombosis with an accuracy of 77% (95% CI 69.9–84.0%) (Figure 1).

The TS score, according to the factors included in the model, is shown in Figure 2. As can be observed, the overall TS score could range between 12 and 30 points, with age ranging between 41 and 70 and D-dimer values ≥ 500 ng/mL, the factors with the highest score values.

Compared with the control group, the median TS score was higher in subjects with thrombosis (29, IQR 28–29 vs. 28, IQR 27–29, *p* = 0.001) (Figure 2). In addition, the frequency of thromboembolisms was proportional to a TS score increase. Thus, a TS score ≥28 had an SE for thrombosis of 88.3% (95% CI 78.7–94.8%) and an NPV of 91% (95% CI 83.2–96%); on the contrary, the SP (29.3%, 95% CI 23.8–35.3%) and PPV (23.8%, 95% CI 18.6–29.8%) were low.

A TS score ≥28 was associated with higher requirements for IMV (74.3% vs. 61.0%, OR 1.8, 95% CI 1.06–3.19, *p* = 0.27) and prone position (70.3% vs. 47.3%, OR 2.6 95% CI 1.52–4.55) compared with subjects with TS score vales ≤28.

Finally, during the hospital stay, 232 doppler ultrasounds were performed, and 47 cases of DVT were identified. We calculated that if a TS score ≥28 was considered before performing these tests, only 178 doppler ultrasounds of the lower limb veins would be indicated, which is a decrease of 23% in the number of tests performed. However, in contrast, only 41 cases of DVT (SE = 87.2%) would be diagnosed.

## 4. Discussion

The high incidence of adverse outcomes associated with thrombosis in COVID-19 individuals highlights the need to develop prediction models to identify patients at higher risk. In our study, subjects with thrombosis experienced worse outcomes, such as more extended hospital and ICU stays, higher rates of sepsis, and increased requirements for IMV, tracheotomy, and prone positions than individuals without thrombosis. Interestingly, the results of our study suggest that the TS score could predict thrombosis in hospitalized COVID-19 individuals within the first 24 h of admission with high sensitivity. In addition, despite the lack of statistical significance in comparing the mortality rates between patients with and without thrombosis a significant association was determined between a TS score ≥28 and IMV and prone position. This finding points to the impact of thromboembolism on the progression and severity of COVID-19 and suggests the possible additional utility of this score to identify subjects at higher risk of worse outcomes.

To the best of our knowledge, this is the first study that developed a clinical prediction rule for thromboembolism in severe COVID-19 patients admitted to an ICU. However, due to the characteristics of our sample, it is still being determined whether the TS score is valid for predicting thrombosis in less severe COVID-19 individuals. Other predictive scores, such as the 3D past score, which was performed in the inpatient COVID-19 population, has a similar sensitivity for thrombosis; however, the rate of ICU-admitted patients was not reported, and its relationship with other adverse outcomes was not studied [16].

As observed in other populations [16,17], D-dimer level elevation would be essential to reach a significant TS score; however, other blood tests must be considered to reach significance. For instance, regardless of age, men with D-dimer elevation must have one or two altered factors to reach a predictive cut-off point. In contrast, women must have two or three other abnormal parameters to reach a TS score ≥28.

Our predictive model found that well-known risk factors for thrombosis and hypercoagulability, such as LDH, fibrinogen, or lymphocyte levels, were not statistically significant [3,18]. However, the relationships between thrombosis and IL6 and CRP, which were included in the score, have been previously explored. For instance, Farouk et al. reported that IL6 levels at admission were related to DVT [19]. Similarly, Smilowitz found that the association between CRP levels and adverse outcomes was consistent in patients with low and high D-dimer levels [20].

Interestingly, our results demonstrated that if the TS score is considered when indicating a doppler ultrasound, the number of tests performed could considerably decrease, which could also decrease the related costs. Nonetheless, it must be noted that in the TS database, the reason for performing this test was not registered; thus, it needs to be clarified whether most of the tests were performed due to clinical suspicion of DVT or due to screening.

Limitations of the study need to be considered. For instance, the TS database includes retrospective records with a significant number of patients with incomplete information. Although we did not impute missing values to build a better model that predicted VTE individually, the exclusion of subjects with >30% of missing data could have led to bias. In addition, we considered the prevalence of thrombosis found in our sample; however, the PPV and the NPV of the score must be adapted to the prevalence of thrombosis in different COVID-19 populations. On the other hand, the blood test results that were considered in the final model were primarily taken in subjects admitted during the first waves of COVID-19, but currently, parameters such as IL6 and CRP are not routinely collected at admission. Thus, the TS score may not be feasible. Finally, this study lacked a validation cohort, so the following steps must include narrow and broad validation of the score in different patient samples and clinical environments that include larger and prospective cohorts.

## 5. Conclusions

The initial evaluation of COVID-19 subjects could play a fundamental role in the early identification of factors predictive for thrombosis. The TS could be an effective tool in clinical decision making for hospitalized COVID-19 population; however, further validation studies must be performed.

## Figures and Tables

**Figure 1 jcm-12-01253-f001:**
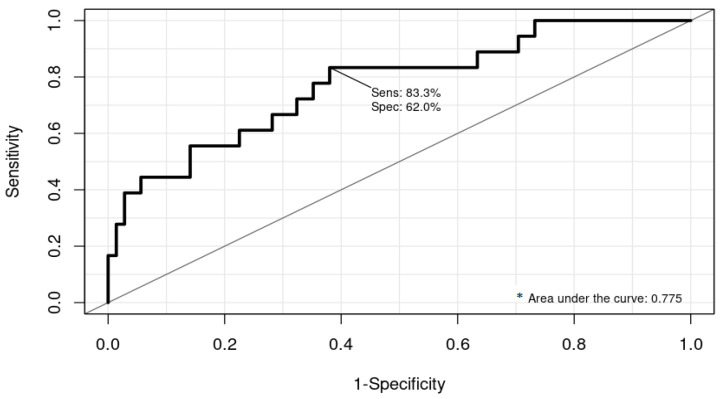
ROC curve and area under the curve (AUC) that assessed the discriminative ability of the final model. * Area under the curve = 0.775; 95% CI 0.6994, 0.8402.

**Figure 2 jcm-12-01253-f002:**
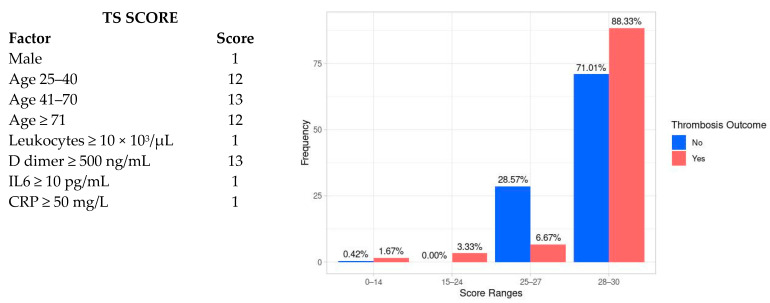
TS score and frequency of thrombosis according to cut-off values of 0–14, 14–24, 25–27, and 28–30.

**Table 1 jcm-12-01253-t001:** Patients’ characteristics, *n* = 299.

Age, Years, Median	62.4 (IQR 51.5–70)
Sex	% (*n*)
Female	28.1 (84)
Male	79.9 (215)
Race	
Caucasian	51.6 (154)
Latin American	11 (33)
Asian	0.3 (1)
African	1 (3)
Arabic	2.7 (8)
Unknown	33.4 (100)
Smoking habit	3.3 (10)
Comorbidities at hospital admission	
Hypertension	40.8 (122)
Diabetes	22.7 (68)
Asthma	3 (9)
COPD	4 (12)
Ischemic heart disease	6.7 (20)
Valvular heart disease	1 (3)
Auricular fibrillation	3 (9)
Obesity	36.1 (108)
Class 1	23.4 (70)
Class 2	8.7 (26)
Class 3	4 (12)
Hospitalization	Median (IQR)
Days from COVID-19 symptoms onset to hospital admission	7 (5–9)
Length of hospital stay, days	28 (18–42)
Time from hospital admission to ICU admission, days	2 (0–5)
Length of ICU stay, days	14 (7–28)
Blood tests results	
D-dimer	1676 (779–4084)
Fibrinogen	719 (608–861)
Leucocytes	8.2 (IQR 5.6–12.4)
Lymphocytes	0.6 (0.4–0.95)
Platelets	201 (147–259)
Ferritin	974 (482.1–1634)
C-reactive protein	126 (69.6–207.6)
Prothrombin time (PT)	12.9 (11.9–15.6)
IL6	71.3 (36.5–167.8)
Creatinine	0.83 (0.68–1.23)
Procalcitonin	0.25 (0.13–0.75)
Lactate dehydrogenase	493.2 (315.5–734.5)
Aspartate dehydrogenase	49 (30.1–80.2)
Alanine transaminase	37.6 (22–71.2)
Doppler ultrasounds of the lower limb veins	77.5 (232)
Thrombosis	20.06 (60)
Deep vein thrombosis (DVT)	10.6 (31)
Pulmonary embolism (PE)	3.67 (11)
DVT + PE	5.01 (15)
Stroke + DVT	0.33 (1)
Stroke	0.66 (2)
Anticoagulant therapy received	% (*n*)
Prophylactic-dose anticoagulation	44.1 (132)
Intermediate-dose anticoagulation	9.03 (27)
Therapeutic-dose anticoagulation	23.4 (70)
Bleeding	5 (15)
Transfusions	
Transfusion of blood components	30.4 (91)
Platelet’s transfusion	7 (21)
Fresh-frozen plasma transfusion	5 (15)
ICU	
Noninvasive mechanical ventilation	58.5 (175)
Invasive mechanical ventilation	70.9 (212)
Tracheotomy	47.2 (141)
Prone positions	59.2 (177)
Sepsis	15.4 (46)
Deaths	29 (87)

**Table 2 jcm-12-01253-t002:** Bivariate analysis between subjects with and without thrombosis.

	Thrombosis *n* = 60 *	No Thrombosis *n* = 239		
	Median (IQR)	Median (IQR)	Median Difference (95% CI)	*p*-Value
Age, years	60 (51–65.9)	63.2 (53–72)	−2.3 (−5.5–0.95)	0.043
Blood test results				
D-dimer	1859.5 (1151–5970)	1605 (772–3335)	280 (−2039.3–2600.8)	0.786
Fibrinogen	813 (567–1020)	781 (625–903)	27.2 (−38.1–92.6)	0.410
Leucocytes	7.85 (5.2–12.1)	7.30 (5.32–10.3)	0.39 (−1.50–0.79)	0.527
Lymphocytes	0.77 (0.47–1.2)	0.70(0.40–1.0)	−0.02 (−0.17–0.10)	0.511
Platelets	239 (173–283)	210 (160–274)	2.6 (−24.9–30.21)	0.851
Ferritin	1006.5 (528–1573.2)	925.4 (474–1634)	138.5 (−624.6–901.8)	0.721
C-reactive protein	120.1 (64.7–277.4)	128.7 (82.4–206.7)	7.32 (−26.5–41.1)	0.668
Prothrombin time	13.1 (12–16.1)	13.4 (11.9–58)	3.5 (−10.9–3.07)	0.268
IL6	95.2 (42–238.2)	71.1 (38–167.8)	10.5 (−73.3–94.4)	0.803
Creatinine	0.86 (0.62–1.01)	0.82 (0.69–1.2)	0.07 (−0.11–0.25)	0.479
Procalcitonin	0.25 (0.14–1.08)	0.22 (0.09–0.53)	−0.045 (−0.137–0.030)	0.303
Lactate dehydrogenase	392 (325–557)	384 (301–559)	−12 (−63.9–39)	0.665
Aspartate dehydrogenase	49.5 (32.5–70.8)	43.5 (29.8–74)	−7.7 (−15.000–13.0)	0.995
Alanine transaminase	43 (29–75)	41 (24.8–72.5)	−3.0 (−17.0–9.0)	0.566
Hospitalization				
Days from COVID-19 onset to hospital admission	6 (4–7)	7 (IQR 5–10)	−1.5 (−3.8–1.1)	0.139
Length of hospital stay, days	35.5 (25–53)	27 (17–37)	10 (2.1–17.9)	0.013
Length of ICU stay	27.5 (15–40)	12 (7–24)	12.8 (5.8–19.9)	0.001
	% (*n*)	% (*n*)	Crude OR (95% CI)	
Gender				
Male	82 (49)	69 (166)	1.95 (0.96–3.98)	0.060
Female	18 (11)	31 (73)		
Race				
Caucasian	56.6 (34)	50.2 (120)		
Latin American	13.3 (8)	10.4 (25)	0.88 (0.36–2.14)	0.819
Lifestyle habits				
Smoker	3.3 (2)	3.3 (8)	0.97 (0.18–4.45)	0.914
Previous comorbidities				
Hypertension	50 (30)	38.4 (92)	1.4 (0.81–2.6)	0.203
Diabetes mellitus	21.6 (13)	23 (55)	0.82 (0.41–1.6)	0.578
Asthma	0 (0)	3.7 (9)	0.79 (0.74–0.84)	0.127
COPD	5 (3)	3.7 (9)	1.3 (0.35–5.1)	0.712
Ischemic heart disease	3.3 (2)	7.5 (18)	0.42 (0.09–1.8)	0.386
Valvular heart disease	1.6 (1)	0.83 (2)	2.0 (0.17–22.5)	0.491
Auricular fibrillation	1.6 (1)	3.3 (8)	0.48 (0.06–3.9)	0.693
Obesity	38.3 (23)	35.1 (84)	0.96 (0.53–1.75)	0.911
Class 1	25 (15)	23 (55)	0.95 (0.48–1.86)	0.894
Class 2	11.6 (7)	7.9 (19)	1.28 (0.50–3.32)	0.598
Class 3	1.6 (1)	4.6 (11)	0.31 (0.04–2.55)	0.257
Bleeding	16.3 (8)	3.8 (7)	4.9 (1.7–14.5)	0.004
Transfusions				
Transfusion of blood components	50 (30)	25.5 (61)	2.9 (1.62–5.23)	0.000
Platelet’s transfusion	13.3 (8)	5.4 (13)	2.6 (1.05–6.78)	0.032
Fresh-frozen plasma transfusion	8.3 (5)	4.1 (10)	2.0 (0.68–6.33)	0.188
ICU management				
Noninvasive mechanical ventilation	60 (36)	58.1 (139)	1.09 (0.60–1.97)	0.764
Invasive mechanical ventilation	88.3 (53)	66.5 (159)	3.8 (1.65–8.76)	0.001
Tracheotomy	60 (36)	43.9 (105)	1.9 (1.06–3.38)	0.028
Prone positions	81.6 (49)	53.5 (128)	3.8 (1.79–8.18)	0.000
Sepsis	33.3 (20)	17.9 (43)	3.06 (1.46–6.39)	0.002
Deaths	38.3 (23)	26.7 (64)	1.70 (0.93–3.07)	0.078

* The cases of thrombosis were distributed as follows: 31 cases of deep vein thrombosis (DVT); 26 of pulmonary embolism (PE) (24 of them peripheric and 2 central PE); 3 cases of stroke. A total of 15 cases of PE and 1 case of stroke also presented DVT. Nine cases of DVT also presented catheter-related thrombosis. No cases of acute myocardial infarction were found in this sample.

## Data Availability

Not applicable.

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
