# Peer review of "A Clinical Prediction Rule for Thrombosis in Critically Ill COVID-19 Patients: Step 1 Results of the Thromcco Study"

_jcm, 2023, doi:10.3390/jcm12041253_

Round 1

Reviewer 1 Report

The authors present a well-written and comprehensible report with a concise but appropriate discussion and limitations section. Most importantly, the limitations of the significant number of patients with incomplete information are discussed. Overall, the methodology appears sound, but requires some clarifications and more detailed descriptions. Specifically, some aspects of the statistical analysis should be reconsidered or explained.

Prevalence impacts the positive predictive value (PPV) and negative predictive value (NPV) of tests. As the prevalence increases, the PPV also increases but the NPV decreases. Similarly, as the prevalence decreases the PPV decreases while the NPV increases. It is necessary to adjust the PPV and NPV values to a different prevalence due to the high sensitivity of PPV and NPV to the prevalence of thrombotic complications in the studied population, as well as its possible bias due to the retrospective nature of the sample and exclusion of patients with >30% of missing data. It is possible to calculate the true prevalence in a cohort of 2055 subjects with COVID-19 registered in the thromcco database.

According to GRADE, it is important to estimate the boundaries of the 95% confidence intervals. For parameters Se, Sp, PPV and NPV, 95% CI must be given.

In the methods section it is necessary to report whether all the quality indicators of the risk prediction model were evaluated in the training set, or in the test set.

Author Response

Reviewer 1

The authors present a well-written and comprehensible report with a concise but appropriate discussion and limitations section. Most importantly, the limitations of the significant number of patients with incomplete information are discussed. Overall, the methodology appears sound, but requires some clarifications and more detailed descriptions. Specifically, some aspects of the statistical analysis should be reconsidered or explained.

Prevalence impacts the positive predictive value (PPV) and negative predictive value (NPV) of tests. As the prevalence increases, the PPV also increases but the NPV decreases. Similarly, as the prevalence decreases the PPV decreases while the NPV increases.

It is necessary to adjust the PPV and NPV values to a different prevalence due to the high sensitivity of PPV and NPV to the prevalence of thrombotic complications in the studied population.

R: The prevalence of thrombosis in the studied population was 20.1%, which was used to calculate the PPV and NPV of our work. The calculations are shown below:

  • PPV = (sensitivity x prevalence) / [ (sensitivity x prevalence) + ((1 – specificity) x (1 – prevalence)) ]
  • PPV = (0.883 x 0.201) / [ ((0.883 x 0.201) + ((1-0.293) x (1-0.201)) ]
  • PPV = (0.883 x 0.201) / [ ((0.883 x 0.201) + (0.707 x 0.799))]
  • PPV = 0.177 / (0.177 + 0.564)
  • PPV =0.177 / 0.741
  • PPV= 0.238
  • PPV = 0.238 x 100 = 8%
  • NPV = (specificity x (1 – prevalence)) / [ (specificity x (1 – prevalence)) + ((1 – sensitivity) x prevalence) ]
  • NPV= (0.293 x (1-0.201)) / [0.293 x (1-0.201) + ((1-0.883) x 0.201)]
  • NPV = 0.293 x 0.799 / [ (0.293 x 0.799) + (0.117 x 0.201)]
  • NPV= 0.234 / [(0.234) + (0.023)]
  • NPV= 0.234 /0.257
  • NPV=910

A statement referring to these calculations has been included in lines 139-142.

Thus, considering the prevalence of thrombosis in our sample (20.1%), we calculated the sensitivity (SE) and specificity (SP) of the score and its respective positive and negative predictive values (PPV and NPV, respectively).”

In addition, in lines 272-274 we have included a statement referring to the importance of adapting the score to the prevalence of thrombosis in different populations.

..we considered the prevalence of thrombosis found in our sample; however, the PPV and the NPV of the score must be adapted to the prevalence of thrombosis in different COVID-19 populations.”

As well as its possible bias due to the retrospective nature of the sample and exclusion of patients with >30% of missing data.

R: In the limitations section we have included a statement referring to the possible bias of excluding patients with missing data. This can be found in lines 270-272 as follows.

Although we did not impute missing values to build a better model that predicted VTE individually, the exclusion of subjects with >30% of missing data could lead to bias."

It is possible to calculate the true prevalence in a cohort of 2055 subjects with COVID-19 registered in the thromcco database.

R: Yes, the prevalence of thrombosis in the entire cohort of 2055 subjects was 22.2% (n=458 cases). This prevalence is similar to the found in our sample and is consistent with the reported in other critically-ill COVID-19 populations.

According to GRADE, it is important to estimate the boundaries of the 95% confidence intervals. For parameters Se, Sp, PPV and NPV, 95% CI must be given.

R: We appreciate that you have noticed this missing information. We have provided the 95% CI of Se, Sp, PPV and NPV in lines 214-216.

“…a TS score ≥28 had a SE for thrombosis of 88.3% (95% CI 78.7%-94.8%) and a NPV of 90.9% (95% CI 83.2%-96%), but on the contrary, the SP (29.3%, 95% CI 23.8%-35.3%) and the PPV (23.9%, 95% CI 18.6%-29.8%) were low.”

In the methods section, it is necessary to report whether all the quality indicators of the risk prediction model were evaluated in the training set, or in the test set.

R: We have included this information in lines 127-130 as follows:

“Once we obtained a model with statistically significant predictors (p-value<0.05) and overall accuracy above 70% (training set), this model was validated through computations of accuracy and performance using the remaining 30% of patients (test set).”

Reviewer 2 Report

This is a study presenting a score for predicting thromboembolic events in patients with severe COVID-19 admitted in ICU. It is a useful and well-written study. Some suggestions for further improvements:

1. The title should mention that the risk score refers to patients with severe Covid-19 and not all hospitalized patients with Covid-19.

2. Table 1: Please define obesity grade 1,2 and 3.

3. Table 1: Why is NYHA score included below the transfusions?

4. Table 1 and Table 2: Table 1 includes 'smoking habit' as a variable, whereas Table 2 'current smoking' in lifestyle habits. Please be consistent.

Author Response

This is a study presenting a score for predicting thromboembolic events in patients with severe COVID-19 admitted in ICU. It is a useful and well-written study. Some suggestions for further improvements:

  1. The title should mention that the risk score refers to patients with severe Covid-19 and not all hospitalized patients with Covid-19.

R: The title has been modified to A clinical prediction rule for thrombosis in critically ill covid-19 patients: step-1 results of the thromcco-study

  1. Table 1: Please define obesity grade 1,2 and 3.

R: The variable obesity has been divided into categories changing the term “grade” for “class”. Thus, based on patients’ body mass index (BMI), the classes were classified as 1, 2, and 3. These definitions have been included in lines 101-102 as follows.

“….obesity [divided into categories as class 1: BMI 30-34.9, class 2: BMI 35-39.9, and class 3: BMI >40]…”

  1. Table 1: Why is NYHA score included below the transfusions?

R: We apologize for this typo; it has been removed.

  1. Table 1 and Table 2: Table 1 includes 'smoking habit' as a variable, whereas Table 2 'current smoking' in lifestyle habits. Please be consistent.

R: To be consistent, the term has been changed to “smoking habit” in both tables.

Reviewer 3 Report

This interesting manuscript describes a clinical prediction rule for hospitalized COVID-19 patients presenting thrombosis-related conditions. The predictive risk model showed a moderate-adequate sensitivity with a moderate-low specificity, with a moderate-adequate AUC that is still limited for clinical implementation (shown in figure 1). 

More importantly, the value of this work is related to the need for predictive models to identify patients at higher risk of thrombosis with COVID-19, which is linked to high worse outcomes and high mortality. It is a first step for developing more predictive models, which should also be done in patients with less severe COVID-19 and associated with the outpatient setting. Along these lines, different predictive models could be built in this and different populations, including different variables and adjusting for critical variables. 

The manuscript contains an adequate English level and requires minimal grammatical revisions.

What was the AUC confidence interval? Please add it to the figure.

For clarity, it would be better if Table 1 is divided between controls and patients and a comparison test between both.

Please include the n. of controls or expand their description.

In Table 2, significant p-values should be highlighted and contain identical decimals (3) for all of them.

Author Response

This interesting manuscript describes a clinical prediction rule for hospitalized COVID-19 patients presenting thrombosis-related conditions. The predictive risk model showed a moderate-adequate sensitivity with a moderate-low specificity, with a moderate-adequate AUC that is still limited for clinical implementation (shown in figure 1).

More importantly, the value of this work is related to the need for predictive models to identify patients at higher risk of thrombosis with COVID-19, which is linked to high worse outcomes and high mortality. It is a first step for developing more predictive models, which should also be done in patients with less severe COVID-19 and associated with the outpatient setting. Along these lines, different predictive models could be built in this and different populations, including different variables and adjusting for critical variables.

The manuscript contains an adequate English level and requires minimal grammatical revisions.

What was the AUC confidence interval? Please add it to the figure.

R: We appreciate noticing this missing information. We have included the 95% CI of the AUC (0.6994, 0.8402) in line 193 and in Figure 1.

For clarity, it would be better if Table 1 is divided between controls and patients and a comparison test between both.

R: This information has been included in Table 2; however, for clarity, the term "control group" has been added.

Please include the n. of controls or expand their description.

R: The definition of the control group can be found in lines 138 and 139 as follows:

To establish predictive cut-off values, subjects that did not present a thrombotic event during hospitalization were considered the control group."

Following your recommendation, we have included the n (n=239) of the control group in line 177.

In Table 2, significant p-values should be highlighted and contain identical decimals (3) for all of them.

R: We have corrected this inconstancy and added three decimals for all p values.
